# Reconstruct(ing) a Hidden History: Black Deaf Canadian Relat(ing) Identity

**Jenelle Rouse** [1,2,*] , **Amelia Palmer** [3] **and Amy Parsons** [4]

1 Faculty of Education, Western University, London, ON N6A 3K7, Canada
2 Centre for Community Services and Early Childhood, George Brown College, Toronto, ON M5R 1M3, Canada
3 Center for Black Deaf Studies, Gallaudet University, Washington, DC 20002, USA
4 Gallaudet Interpreting Service, Gallaudet University, Washington, DC 20002, USA
* Correspondence: research.blackdeafcanada@gmail.com

**Abstract:** Black Deaf Canadians are under-represented in every facet of life. Black Deaf Canadian excellence, history, culture, and language are under-documented and under-reported. *Where are we in history? Where are we now? Why are we not being documented?* Black Deaf Canada was established to address these long-standing issues and went on to create an independent research team that led a project called "Black Deaf History in Canada". This article provides an early account of how the community-based research team conducted a relationship-building practice prior to and during a three-week research trip. Black Deaf Canadians' relat(ing) experience in history has inspired us to fight for inclusivity.

**Keywords:** Black Deaf Canadian; identity; relationship-building; sign language; reworlding

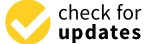



## 1. Introduction

The year 2020 is when everything in our lives shifted: the global pandemic; lockdown; social and physical distancing; job loss. Lives were senselessly lost—specifically, the life of George Floyd. A series of losses and chaos provoked many Black Deaf Canadians to reach out for a safe space to obtain comfort, support, and resilience. Natasha "Courage" Bacchus, a Black Deaf Canadian athlete turned performer, decided to reach out and host a private group talk to reflect on the idea of a Deaf Black Lives Matter (Deaf BLM) movement. Through social networking and online communication, Bacchus invited Black Deaf Canadian leaders of different professions—such as teachers, artists, athletes, and business owners—to come together for regular online sessions over several months to share individual experiences. During these sessions, our upbringings, identities, languages, cultures, and education were shared amongst the group.

There was a sense of understanding and belonging within the group. Black Deaf Canadians live in predominantly white Deaf communities, separated by long distances within a large country, making it quite challenging to find each other. While we live in various provinces and territories, we have a shared experience of isolation in the face of abnormal circumstances. Despite each member of the group identifying as Black and Deaf (what we term "Black Deaf"[1]), our understanding of, and access to, Black Deaf Canadian scholarship is challenged by audist attitudes and practices around Sign Language and the Deaf community. Audism describes audiocentric, audiosupremacist assumptions and attitudes around hearing, speaking, and behaving (Humphries 1977). Several questions emerged from the Canadian Deaf Black Lives Matter (BLM) group: Where are we in history? Where are we now? Why are we not being documented? These questions led to the formation of the research project "Black Deaf History in Canada". This research project explores these questions through a theoretical framework that brings together the concepts of Black Deaf Gain (Moges 2020) and DisCrit: Disability Studies and Critical Race Theory (Annamma et al. 2016).

Rezenet Tsegay Moges' (2020) seminal work on Black Deaf identity theorizes Black Deaf Gain as an unintended consequence of segregation and the use of Sign Language as a vehicle towards positive identity formation among Black Deaf individuals in educational contexts (From White Deaf People's Adversity to Black Deaf Gain, pp. 81–84). Moges' work is a critical counternarrative response to Bauman and Murray's (2009, 2014) coining of the initial assent term "Deaf Gain" that counters the deficit frame of terms such as "hearing loss". "Deaf Gain" refers to "unique cognitive, creative, and cultural gains manifested through deaf ways of being in the world" (Bauman and Murray 2014, pp. xv, 15). Moges (2020) built on Bauman and Murray's (2009, 2014) work to further reframe ontological notions of Deaf identity—for example, regaining a Sign Language that was historically excluded as well as studying "new" linguistic growth in Sign Languages at different ages, geographic spaces, and/or stages of cognitive development. Moges (2020) proposed that Black Deaf Gain, overlooked by Bauman and Murray (2009, 2014), needs to be acknowledged. Additionally, Moges (2020) focused on the social and historical experiences of Black Deaf people and Black Sign Language as being worthy of celebration and preservation.

In the United States, Black American Sign Language (in this case, Black ASL) has since been preserved. According to McCaskill et al. (2011), Black ASL is perceived as distinct from the use of American Sign Language (ASL) by white individuals. This is because the vernacular of African American English (AAE) is derived from middle-class white English people. Social and historical experiences of Black Deaf people are featured in Dr. Carolyn McCaskill et al.'s (2011) sociolinguistic research, as observed in historic literature on racial segregation and Civil War America, including racial desegregation in the South (after cases such as Brown vs. the Board of Education).[2] McCaskill et al. recognized, identified, and understood the differences between Black Deaf individuals and the social conditions that impact the emergence of various Sign Languages. Other than the differences between Black and white use of their respective Sign Languages, there are some identified varieties with the Black Deaf community, such as locations, ages, segregated deaf schools, integrated deaf schools, mouth movements, and education. McCaskill and her research team (McCaskill et al. 2011, 2022; McCaskill and Hill 2016) confirmed that signing while Black is evidenced in natural, traditional Sign Languages with distinct phonological, syntax, and grammatical structures and cultural, regional, and linguistic features, as found in policies and reports, some dictionaries, and ASL teaching classes.

Although the author and contributing writers ("authors" going forward) of this article appreciate these facts, a Canadian perspective is needed to determine whether Black Deaf Canadians who prefer Sign Language similarly benefit from history shaping their "language, culture, and identities" (Moges 2020, p. 73). Although the authors of this article concur with McCaskill et al.'s (2011) point that there are several geographic and social factors involved in the formation of language varieties (from a sociolinguistic perspective), the benefits and historical preservation of Black ASL within communities are different between Canada and the United States. Black ASL is influenced differently by, for example, Plains Indian Sign Languages (lingua franca), Caribbean Creole Languages, African Languages, French, and English. As proposed by McCaskill et al. (2011), the changes within Canadian Black Sign Languages are affected by upbringing, language contact, and social interactions and norms that, in turn, influence the understanding of identifying as a Black Deaf Canadian who uses Sign Language.

While adopting Moges' (2020) Black Deaf Gain theory with an embedded Canadian perspective, the authors observed Annamma et al.'s (2016) combined theories of Disability Studies and Critical Race (or DisCrit). DisCrit has seven foundational tenets, wherein it (1) observes the ways ableism, racism, (and audism) demonstrate interdependence and behavior that upholds a standard of "normalcy"; (2) celebrates intersectional or multi-dimensional identities and rejects either/or notions of identity, such as race, dis/ability, class, gender, and sexuality; (3) recognizes Western cultural norms of labeling social constructions of race and ability; (4) unveils invisible narratives of marginalized populations that are not traditionally acknowledged within research; (5) thinks about how legal and

historical information (e.g., policies) un/intentionally deny rights to some citizens based on dis/ability and race; (6) notices a variety of superficial relations that racialized people with disabilities have to endure; and (7) requires activism and supports all forms of resistance (Annamma et al. 2016). Ultimately, each tenet serves to strengthen a theoretical framework for a radically intersectional approach to the topic of race and disability in institutional and community contexts. The authors of this article focus on tenets three through six as they are most relevant to this study.

The combined theories of Black Deaf Gain and DisCrit with lived Canadian perspective deliver continuous growth in societal, artistic, and academic areas to "branch out, build, and increase our understanding by considering different sides and exploring various perspectives to find deeper insights or greater truths than presently exist" (Moges 2020, p. 94). As such, "Black Deaf History in Canada" is a significant contribution to a growing body of narratives that were previously neglected. Furthermore, the present research—along with historical information and lived experiences of Black Deaf Canadians—will inspire a paradigm shift in multidisciplinary academic-, professional-, and community-based fields, such as Black Deaf Studies, Black Disability, Black History, Black Arts, Deaf Education studies, and so on. This practice of genuine community-building relationships will encourage us to rethink how we, as Black individuals, relate to our identity. It will lead to a more precise identity (re)definition with full transparency and communication and easier partnerships and community building.

This article provides an account of how establishing our organization enabled the authors, as community-based researchers, to build their individual and collective personal and professional relationships and conduct a research project. This article also shares the authors' commitment to restore and (re)construct histories of Black Deaf Canadians and outlines their research methods at various research sites (archives and museums). New insights that emerged at the early stage of data collection are presented next. This is followed by the concluding considerations of how vital community-building engagement is for incidental or direct learning to ensure that narratives of Black Deaf Canadians are identifiable, tangible, and relatable.

Throughout this article, the authors argue that Black Deaf/Black disabled Canadians are excluded from Canadian history as a result of layered oppression. Black Deaf Canadian narratives are identified as under-documented/under-represented. The processes of relationship-building and decision-making emerging from the research project and (re)constructing history with the Black community lead to the redefinition of Black Deaf Canadian relatable identity/ies.

## 2. Under-Documentation and Under-Representation of Black Deaf Canadians[3]

Our literature review into the work of scholars focusing on Black disabilities primarily revolves around three major themes: (a) the theory and praxis of critical race and/or critical disabilities within education (McCaskill et al. 2011; Stapleton 2015); (b) the acquired disability experience of Black Americans in later life (Harder et al. 2019); and (c) contrasting the Black disabled experience with the white disabled experience—through a white lens (Moges 2020; Renken et al. 2021). Lived experiences of cultural identities are often overlooked in various traditional research paradigms and bodies of literature. It remains imperative that Deaf researchers from their aligned cultural groups have the opportunity and access to explore, identify, and name how and in what ways our respective Black communities embrace and accept us as Deaf beings.

Deaf identities within the dominant "hearing society" face oppression, as evidenced by white American author Tom Humphries (1977), who identified its root cause of audism.[4] For him, audism means that "one is superior based on one's ability to hear or behave in the manner of one who hears. It is being understood as the bias and prejudice of hearing people against deaf people" (Humphries 1977, pp. 12–13). Over time, the term has gained more nuanced and complex meanings relevant to the authors' experiences. These definitions range across multiple contexts, such as individual—an assumption that

Deaf people cannot live independently (Gertz 2008; Humphries 1977; Lane [1992] 1999); institutional—making authoritative decisions about Deaf people's lives without their engagement (Bauman 2004; Lane [1992] 1999); metaphysical—attitudinal prioritization of speaking over signing (Bauman 2004); and finally, an incorporated critical race perspective regarding laissez-faire prejudices—recognizing and dismissing the existence of Deaf people with intersectional identities (Eckert 2010).

Cho et al. (2013, p. 786) discussed the meaning of external and internal layered and intersectional oppression within audiocentric Black communities, defining "the multiple ways that race and gender interact with class in the labor market" and "interrogating the ways that states constitute regulatory regimes of identity, reproduction, and family formation" to determine "whether the subject is statically situated in terms of identity, geography, or temporality". Oppression is intersectional given the instances where race and gender are noticeably discussed, yet the experiences of race, gender, and disability are often excluded.

One example of this occurs when discussing the experience of Viola Desmond—a Black Canadian woman—who was arrested for sitting in a whites-only section at a theatre in the middle of the 20th century. She had to negotiate for her preferred seating and consequently became a civil rights activist: "... I told [the movie theatre manager] that I never sit upstairs because I can't see very well from that distance" (Backhouse 2017, p. 103). The significance of this reasoning is often not included in contemporary analyses of her story. If the previously mentioned statement is closely and carefully considered, it would reveal the possibility that Desmond had a disability (i.e., low vision). While she experienced racial discrimination, her accessibility needs as a person with a disability were also denied. At that time, fundamental human rights and disability rights were apparently not Canada's top priority when individuals with disabilities put forth their experiences of such barriers. It is our observation that Black disability/critical disability academic research is largely reserved for chronic, noncongenital medical conditions such as heart disease, diabetes, obesity, and so on. Based on this observation, Black Canadians are often considered to be Black or not Black when discussing the experiences of oppression and/or barriers.

In another example, a discussion of how race has become a more insidious aspect of the African Canadian experience in Carl James et al.'s (2010) Race and Well-Being: The Lives, Hopes and Activism of African Canadians, there is scant mention of disability or Deaf ways of being in the entire, otherwise essential, text. Black Deaf Canadian identity is clearly under-represented in academic research and in Canadian contexts; being Deaf or disabled becomes buried and invisible under layers of oppression.

When considering the identities held by Black Deaf Canadians, especially those who prefer Sign Language, the experience of audism is conflated with racism. For example, while Eckert (2010) acknowledged racism and audism within Deaf identity studies, the study was largely conducted through a white lens. This can also be seen in Eckert and Rowley (2013), Humphries (1977), and Ladd's (2003) seminal Deafhood work. These works serve as a critical praxis for understanding how white (audio-centric) supremacy is designed to prevent and/or eradicate nonwhite Deaf ways of being within educational, societal, and medical contexts; the latest frameworks serve as the de facto ways of interpreting the Deaf experience, the majority of them operating within a white lens.

Deaf academic research is overwhelmingly white. Research participants are also overwhelmingly white. Participants either have ties to Deaf schools or multigenerational Deaf families where there were no barriers to the language of access—Sign Language. Deaf scholarship, as a burgeoning field, lies mainly in the hands of Deaf elites—those with higher levels of education, from the middle class, and/or from strong Deaf ecosystem networks. The parallel discussions, research, and analysis of Deaf and Black identities are often born from an either/or approach: Black or Deaf. It was not until very recently that discussions of the Black Deaf experience emerged in academic work, such as Chapple (2019), McCaskill and Hill (2016), McCaskill et al. (2011), and Stapleton (2015). Respectively, these works discuss the Black Deaf female experience, Black Sign Language, the experience of Black

Deaf learners, and the process of working with Black Deaf participants. Black Deaf Canada is interested in the Black Deaf Canadian experience as a holistic and encompassing area of research.

It is crucial to make an explicit distinction between audism and ableism—the former focuses on the Deaf experience of moving about in a world that is audiocentric. In contrast, the latter focuses on the experience of the disabled as enforced by the abled society. Scholarship focusing on ableism in Black hearing contexts is scarce, adopting a deficit rather than celebratory mindset. Chapple (2019) reinforced this point by positing a Black Deaf feminist framework that is inclusive of disability analysis; however, her work is rooted in the feminist and female experience, whereas we are more interested in a broader systematic analysis of the Black Deaf Canadian experience. The literature captures systemic oppression that has continued to work against Black individuals and communities to create an "othering" of the Black Deaf experience (McCaskill and Hill 2016; McCaskill et al. 2011; Stapleton 2015). Harder et al. (2019) discussed ableism in communities where they found that implicit (indirect) prejudice against people with disabilities was strongest among Black and male participants. This type of prejudice has increased in older segments of the community (see Harder et al. 2019; see also Gillborn 2015).

Language shaming is another example of prejudice affected by ableism and audism when Black Deaf history is not accounted for in Canadian studies. Language shaming, the enactment of language subordination, is further defined at an institutional/governmental level with respect to Sign Language (Haualand and Holmstrom 2019; Piller 2017). Language shaming is a definite, tangible experience for Black Deaf Sign Language users, of whom there are few. Although Sign Language has been identified as a language in its own right for decades (see Stokoe 1960), it is very much an othered experience within Black communities. Piller (2017, n.p.) argued that "like other forms of stigma, language shame may have deleterious effects on the groups and individuals concerned and may result in low self-esteem, a lack of self-worth, and social alienation".

The research team recognizes that the terms "Deaf" and/or "Black Deaf" are being excluded from Black audiocentric communities. For example, a survey conducted during a Juneteenth event hosted by the DC Area Black Deaf Advocates revealed that 42 percent of respondents—all of whom were Black Deaf—concurred strongly with the phrase, "*My experience with ASL in the Black hearing community has been somewhat negative*".[5] In reference to the experience of language shaming, a research team member shared their recollection of the range of public visceral reactions when signing in American Sign Language (ASL) with peers:

> Black Deaf friends and colleagues and I often remark on the uncomfortable experience of communicating and using ASL in Black hearing-dominant spaces. We are often infantilized, dismissed, and/or blessed by our elders. On the streets among our youth, we are on the receiving end of knee-jerk responses for the possibility of using gang signs; dismissed once it becomes evident that our language is just that—language. (Parsons 2022a)

By identifying and exploring the intersectional experiences of Black Canadians with congenital and noncongenital disabilities, specifically Black Sign Language users, we hope to name, spotlight, discuss, and ultimately contribute to an expanded body of work around the lives of Black Deaf Canadians. Racist and ableist stigmatizing attitudes and practices around Sign Language extend beyond the team's own individuality and even beyond their Black Deaf community.

With that in mind, contributions of and to all facets of the lives of Black Canadians have been explored, discussed, and defended since the beginning of the 19th century (e.g., Clifford [1999] 2006; Thomas 2021; Walker 2019). However, as we have outlined, Black Deaf Canadians and those with a disability are rarely discussed in conversations or publications. In response to this absence, two Black Canadian individuals—one Deaf artist, Tamkya Bullen, and one hearing activist[6]—connected and began to advocate together for social justice, creating a club that eventually became our organization, currently known as Black Deaf

Canada. This enabled alternative incidental learning about identities—in particular, history, culture, and language that emerged during the community-based implementation process.

In terms of the implementation process, there are a handful of scholarly sources on Deaf research team building or experiences, including Deaf and hearing team partnerships. In this instance, Renken et al. (2021) conducted a case study on the identity development of three Black Deaf secondary students during their summer course of Science, Technology, Engineering, and Math (STEM). From photographic observations, the researchers learned that students' individual identities were being defined and shaped by social and academic engagements. Peer mentorship—a form of relationship building—reveals sensitive yet complicated interactions in which students experience a sense of racial under-representation that impacts their identity development and decision-making. Student identity development often comes from self-analysis, leading to decisions on whether they identify with the term Black Deaf individual. Renken et al. (2021, p. 1109) referred to Schmitt and Leigh's (2015) literature-related observation that Deaf culture is "based upon white Deaf culture and that Black Deaf individuals may perceive themselves as doubly marginalized, from Deaf culture because of their race, and from Black culture because of their deafness".

Wolsey et al. (2017) explored the importance of clear communication within a partnership between Deaf and hearing professionals. According to the authors, vulnerability occurs through the practice of reflecting on one's power/powerless position concerning their audiological status and Sign Language awareness. The Deaf–hearing relationship takes a transformative form that offers the team guidance in comprehending and enriching their knowledge/skills. The authors add that, in spite of their similar academic statuses, the vulnerability of the power imbalance heightens the level of sensitivity in their relationship building by acknowledging their strengths and areas of need.

These examples reveal that many internalized perceptions impact our identity development. We have observed and discussed ableism, audism, racism, and language shaming. With these forces in mind, lived experiences of Black Deaf Canadians who use Sign Language are typically under-documented and under-represented. In the following section, we look at how Bullen's club, the "Black Deaf Club of Canada", initiated and transformed into Black Deaf Canada.

## 3. From the Black Deaf Club of Canada to Black Deaf Canada

On the nuances of Deaf identity, Leigh et al. (Leigh et al. [1992] 2020, p. 155) collaboratively wrote

> When we ask [Deaf] individuals to describe themselves, they may accept or not accept calling themselves deaf or Deaf, depending on their situation or experience. This can change over time. Those who identify themselves as culturally Deaf are individuals who use . . . signed language . . . , feel strongly that being Deaf is just fine or a gain, socialize with and get support from other culturally Deaf persons, and live a 'Deaf' way of life. They feel at home with each other.

Being "at home with each other" could describe the feeling at the core of Bullen's club that was formed in 2016. It began when Bullen met the activist (who prefers to remain anonymous) at a community activity. They shared the concern that Black Deaf and Black Deaf individuals with disabilities in Canada were often not being fully included in areas such as mental health resources, mindfulness workshops, and accessibility. The activist and Bullen observed that a Deaf-led club should be formed. Bullen and the activist developed a mission statement and publicly posted the statement on a Facebook page. The name of the club, "Black Deaf Club of Canada" was created.

The Facebook page, "Black Deaf Club of Canada", attracted the attention of Abigail Danquah, a then Canadian college student living in the United States. When Danquah saw the club's statement encouraging Black Deaf Canadians to exchange knowledge, experience, and support, she immediately contacted Bullen to request a meeting. Bullen and Danquah bonded through a series of discussions, which led them to create a safe space in which more individuals could participate.

Bullen and Danquah started small by inviting two Black Deaf Canadians, Bacchus and Andrea Zackary (business owner) to a familiar meeting place at a university. At the campus, Bacchus, Bullen, Danquah, and Zackary met to discuss ideas of how to get Bullen's club recognized on a public scale. After a few gatherings, Danquah and Zackary volunteered to help Bullen to lead the club. They debated how the club could be best established or structured in order to get financial support to afford the resources that Bullen dreamed of. However, Bullen, Danquah, and Zackary had different visions that have yet to be realized cohesively. They eventually chose to keep the club as is, without financial support.

The club operated for two years in a safe space with a few activities (e.g., workshops, social events, and drop-in gatherings). Bullen and Danquah reconnected online in August 2020 to discuss ways to continue the club. They then reached out to a Deaf BLM online group with a request for support in moving the club's initiatives forward. Three individuals answered the call, and the group worked on reviewing the club's mission to ensure that it was clear and achievable. As a result, the "Black Deaf Club of Canada" turned into the community-based project "Black Deaf Canada". Since then, Black Deaf Canada has become primarily a social network space where individuals of all ages can find each other online. The name may be subject to change in the future after a series of research and community-based consultations.

## 4. Research Team: Relationship-Building and Decision-Making Process

Black Deaf Canada empowers Black Deaf Canadian individuals by engaging, networking, collaborating, and forming community-building practices while resisting ableism, audism, and subtle racism. Black Deaf Canada has the skills and ability to influence changes in social and material circumstances, a practice which takes time. Team building occurs through a shared interest in a project that acknowledges members' human experiences and socioeconomic realities. All members of the research team share some similar experiences, as observed through trust, relationship, and identity development. For example, while our upbringings, practices, and behaviors toward Black Deaf, Black hearing, and non-Black individuals are different, we chose to engage in a series of much needed dialogues together that led us to eventually appreciate who and what we are.

To ensure the effectiveness of this project, relationships with scholars, educators, artists, and the Sign Language community are maintained while also recruiting highly driven Black Deaf Canadian individuals who want to pursue independent studies of Black Deaf Canadian history. At least three research team members were naturally recruited because of their interest in sharing their preliminary findings with the public. Team member/author biographies follow as a means to better elaborate on individual team members' respective and shared motivation in coming together to do this work. These short biographical accounts demonstrate some of our decision-making processes that helped us arrive to the point of adopting critical, transformative race theory as our methodology.

Abigail Danquah:

*I was the one who saw the original founder's club posted online and became a co-founder. I identified as a Black person and later identified as Black Deaf. I moved to Canada for schooling and employment opportunities. With a degree in Applied Arts and Science (B.Sc.) specializing in advertising, marketing, communication, and public relations, including Deaf Cultural Studies, I have an interest in finding statistical information on Black Deaf Canadian Population.*

Dr. Jenelle Rouse:

*I am Black Deaf, Canadian-born, and raised in a hearing Caribbean, African-diaspora family. I am an experienced educator, scholar, adjunct professor, translator, consultant, presenter, and artist. In the fields of traditional and non-traditional education, I have been working with students of various grades, teacher candidates, interpreter candidates, and other professionals. As a listener and writer, I have an interest in sharing knowledge and experience with the public—academic and non-academic alike. In 2017, Danquah reached out to me during my studies, asking for my contribution to formative plans concerning*

*Sign Language and education. Although the timing was not a right fit in my schedule at that time, I expressed an interest in participating in turning the vision of the Black Deaf Canada into a reality after the completion of my studies. Once my studies were completed in the summer of 2020, I immediately joined Black Deaf Canada as a consultant and eventually became a co-director in 2021. Danquah teamed up with me to undertake the community-based interdisciplinary Black, Deaf-led Canadian research project, "Black Deaf History in Canada".*

Amelia Palmer:

*I am a history enthusiast, a community researcher, Black Deaf Canadian-born, and raised in my Caribbean family. At the time of writing, I am currently an undergraduate and paraprofessional student majoring in Deaf Studies and minoring in Linguistics. I have a keen interest in researching, observing, and documenting the linguistic perspective of Canadian Black ASL or Canadian Black Sign Language (CBSL), including Black Deaf History, with hidden lineage stories of Black Deaf Canadian individuals. While working at the university as a paraprofessional student, I noticed that the majority of Black Deaf individuals in North America are not aware of their relat(ing)[7] history during the 18th and 19th centuries. I feel the need to amplify a Black Deaf Canadian narrative as it is part of North American history.*

Amy Parsons:

*I am a Black deaf woman raised by two loving parents in K'jipuktuk, Turtle Island (Halifax, Nova Scotia, Canada). My roots run deep in the African Nova Scotian and Irish heritage communities. My familial roots originate from the Black communities of Lucasville and Weymouth Falls. As an activist for language, educational and economic justice, I work to dismantle hegemonies within educational systems and communities. I bring a historical perspective and scholarly analysis to the constructs of identity, disability, ableism, othering and equity in various racialized communities, particularly in Canadian contexts. Identifying as a queer woman, I continue to unpack and unlearn the effects of decades of experiencing an interpreted education through whiteness.*

Simone Edwards-Forde:

*I was born and raised on an island until I was three years old. Once I moved to Canada, I looked for connections with Black Deaf persons. I was also looking for information about Black Deaf history in my workplace where I am an educator. For years, I found nothing. I become an active community member of Black Deaf Canada where I learned about two main events in 2020 and 2021: panel discussions. The panel presenters were all Black and Canadian. During the second panel discussion's question and answer session, I assertively asked the research panelists if I could join their organization in order to learn more. Since January 2022, I have been an active member of the research project as a research assistant. I feel connected to this team. The atmosphere is warm, kind, respectful, and open to discussing raw feelings without a sense of dismissal, exemplifying the true meaning of teamwork.*

## 5. Emergence of the Research Project

While the members of the research team have different upbringings, interests and motivations and live in different regions across North America, it is apparent that we came together for similar reasons. For more than a year, we met periodically via Zoom meetings and the WhatsApp platform for individual and group study sessions. At the beginning of our relationship-building period, we faced one of the key challenges: vulnerability.

Vulnerability is, as Judith Butler (2001, 2012) described, a physical and conscious act of recognizing and understanding our responsibility to identify what binds us (in terms of identity). Vulnerability also comprises interdependence, meaning we find our commonalities through specific "marks of national, cultural, religious, racial belonging" (Butler 2012, pp. 135–36). Our group is bound together by our shared characteristics, such as our nation (born and raised in or moved to Canada), culture (Sign Language, Deaf), and

race (Black). However, "the bounded and living appearance of the body is the condition of being exposed to the other, exposed to solicitation, education, passion, injury, exposed in ways that sustain us but also in ways that can destroy us" (Butler 2012, p. 141). We may give our lives over, in each other's hands, and/or at each other's mercy. In a sense, vulnerability is not necessarily susceptible to violence, it is a form of relationality. It is an additional practice of developing relationships between individuals towards a sense of solidarity through, for example, actions of cultural conditioning and shaping.[8]

Once again, vulnerability, in our case, refers to our identities in terms of race, Deafness, and, for some of us, gender. We reframe vulnerability as a type of power when critically thinking about how we use specific language to portray our experience and understanding towards historical and present contexts (See Carbin 1996; for critical engagement as per collective and self-learning, see hooks 1994, 2010, 2013). While there is no adequate way of preparing to maintain control over how we share our lives with each other, our ethical relations are something we must address. In our discussions with one another, we learned that at different times we have each asked ourselves: *Do I disclose critical information or keep it to myself?* The asking of this question shows that we did not know each other well as we would have liked because we habitually practiced individualism. We opened up to one another. We developed a shared bond, respect, and understanding that led to a motivation to further each other's innate capabilities and interests (Butler 2001; Pethebridge 2016).

To support ourselves in getting to know one another and the development of our collective sense of identity in our newfound organization, we started with an "on pulse" exercise at the beginning of every online meeting, a technique inviting us to share about our lives and how we are doing. We then proceeded to discuss what we had learned from our conversations and make connections between our research interests, skills, and knowledge. Ongoing dialogue and investigation as a team influenced the process of reconstructing Black Deaf Canadian identity. Acceptance of individual vulnerability, from the moment we first had a meeting right through working together as a team, eventually allowed us to build a strong sense of collectivity. We knew that we could embrace uncertainty, because we trusted each other to be authentic and transparent.

We continued to meet and build relationships through a recruitment process that, in turn, led us to make decisions regarding our research project, "Black Deaf History in Canada". This project comes from a decision-making practice that led the team to consider the intersections in the literature between Black Deaf perspectives and vulnerability. Dr. Samantha Schalk (2014, p. 25) pushed for a better future by ensuring that all Black-related literature features, narratives, and perspectives of Black people with disabilities are acknowledged: "For marginalized people, [speculation] can mean imagining a future or alternative space where one's oppression no longer occurs or in which relations between currently empowered and disempowered groups are incredibly improved". To consciously move away from various oppressions (e.g., racism, ableism, audism), we explored different ways of presenting history and current Black Deaf narratives in a Canadian context that are iterative and attentive to our individual and collective identity. From our meetings, the following questions arose: *Where are we in history? Where are we now? Why are we not being documented?*

To address these questions, the research team voluntarily pursued a much-needed research project on a limited budget to begin and continue to amplify historical and current narratives of Black Deaf individuals. At the time of the COVID-19 lockdowns and social/physical distancing, it was difficult for the team to find specific information online that addressed both Black and Deaf/disabled experiences. It is crucial for the research team to reframe history and current narratives by getting accurate information on documented Black and Deaf/disabled Canadian individuals.

The following section presents the detailed procedures used for our data collection. Some details of our dialogue are also provided to give the reader a clear picture of the data-gathering process. The sections that follow briefly introduce our first research trip and interactions with the community, which supported us in locating Black and Deaf Canadians. Preliminary findings are highlighted, and we discuss how this scholarly experi-

ence informs a historical understanding of social identity (re)construction and provides deeper insight into how community-building practice impacts our identity development and shared history.

## 6. Method

Individual and collective identity development can occur when uncovering and discussing history. Although new to archival methodology, we chose to visit libraries, museums, and archives to read, reflect, discuss, and document using our exploratory approach. This exploratory method is a learning process in which we visualize, develop, and strengthen our research knowledge and skills in tracing sources. While we each have different skills and experiences, we agreed that this combined methodology of Black Deaf Gain and DisCrit is a good start for us to pinpoint significant pieces of history that allow us to redefine the meaning of Black History in Canadian contexts. Winston (2021, p. 293) suggested that we embrace "counter stories and alternate ways of knowing and performing archival work, particularly when these ways of knowing come from groups historically oppressed". This type of work requires time and effort.

Our anticipated long-term research field work was deferred to July 2022 due to the consequences of extended lockdowns and other personal/professional obligations. When we began the early stages of data collection at our own pace, multiple sources of data led us to the Atlantic provinces to visit different research sites and engage in conversations with three individual participants.

### 6.1. Research Sites: Libraries, Museums, and Archives

As Black Deaf Canada maintains a small collection of digital information (videos, photos), the research team gradually added to the documents we had collected prior to our research trip by choosing to visit Nova Scotia, New Brunswick, and Prince Edward Island. The team also developed clear and simple guidelines to follow when conducting conversations (see Figure 1 below).

The process of selecting participants and locations for conversations in Nova Scotia will be discussed in later sections. We started with community-building during our research trip. To practice community-building, everyone on the team was able to either take turns or work together to employ research methods and exploratory tools. We travelled together as a team to the Atlantic provinces, communicated our needs and wants to one another, found and shared information, and had in-depth conversations with participants). The different places we visited for research were as follows: (1) a self-tour of a street housing an established school for the deaf. (2) libraries. (3) archives, and (4) museums. Some of the sites were accessible to us because they provided qualified Sign Language interpreters who were supportive of our research and open to collaborations. We visited these areas to meet people, share, and learn from each other's historical and contemporary stories. Through touring and researching different areas over a three-week visit, we were able to confirm and strengthen prior knowledge gained from conversations, visits, and literature reviews.

In terms of the archival research process, we chose to use an open-ended exploratory approach. We started with libraries and then museums to see what we could learn from books, maps, and photos about where Black Deaf individuals might have been in the 19th century. We found that the sources directed us to specific archival sites for primary sources (newspaper articles, photos, and records) that may contain traces of Black Deaf local individuals of all ages, such as school records and service organizations. With the support of individuals who worked at the archival sites, we were able to compare and validate a list of hard and digital copies of history books, maps, newspaper articles, and photos before categorizing and sorting them.

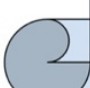

## Preparation Guideline

Make sure both parties (members of the research team and consented participants) are comfortable and understand that:

- Camcorders are on at all the time;
- Contribution to conversational data is voluntary and can withdraw at any time;
- New information after withdrawal will not be added unless consented

**Conservation Starter(s):**
1. Always make sure a consented participant signs a consent form before starting a conversation.
2. For social media purpose, ask for a verbal permission to record a separate video sharing some "sneak-peek" at, for example, artwork, social gatherings, meals, or other that may not be likely shared in print
3. At the beginning of the conversation, start with easy questions, such as their name, where and when born, names of family members
4. Allow the participant to do the talking and ask them to elaborate specific information (e.g. "Tell me more about…" "What do you remember about…")

**Video-tape Setting:**
- Two camcorders
  - One on participant(s)
  - Another on both conversation starters and participants (the space where they are positioned)
- Personal electronic devices (e.g., wireless phones) - Live or Story in social media

**Figure 1.** These Preparation Guidelines are a helpful tool for our research members to follow through prior to and during conversations with participants.

Finding specific information on Black Deaf Canadians in archival spaces using a genealogical approach is painfully challenging. Most names were not chronologically dated, nor were they distinguished as Black or non-Black in, for example, an archival handwritten census. Although we took a variety of different steps in finding information for the project, all actions required time, patience, and focus. Based on this experience, we began to develop archival methods that made sense to us. Visiting and reading history books at libraries was one of our initial steps that helped to narrow the focus. Books, such as Lawrence Hill's (2007) *The Book of Negroes* and Jim Hornby's (1991) *Black Islanders: Prince Edward Island's Historical Black Community*, offer keywords that afford parallel clues to potentially Black Deaf individuals' locations, family names, and timelines.

For example, one member of our team found maps in an atlas in the African History and Culture section at the Nova Scotia Library. The team compared the maps—one modern and another tracing back to the 19th century—and reviewed related books. This activity allowed us to identify and record the names of streets that were mentioned in the books. This information was then manually documented in our database that we refer to for review, analysis, speculation, and iterative processes about how we want to use the information. In the next step, we visited museums, such as Africville Museum, New Brunswick Black History Society, and Prince Edward Island's private museum, where we were able to validate whether secondary sources were similar to what we had documented in the database. Once confirmed, we contacted archivists with a list of Black Deaf Canadians and accurate information from our research.

Developing friendly relationships and a good rapport with genealogists and experienced archivists were key to help reduce the work from months to weeks to days. The research experience is enriched by clearly explaining specific information concerning Black Deaf Canadian history to genealogists/archivists. Presenting and exchanging educational

information between researchers and archivists/genealogists further inspires meaning in relationship and community-building, wherein knowledge is openly shared, developed, documented, and valued. Although this archival research is still at an early stage, our experience helped to narrow down the specific questions that we wanted to ask three community-based participants.

### 6.2. Participants

On the trip, the team used a snowball sampling approach to recruit Black Deaf persons with whom to talk. According to Goodman (2011), snowball sampling is an appropriate technique to use in a study such as ours, since Black Deaf Canadian individuals are foreseeably hard to reach. This snowball technique has been a consistent tool in this community-based research project, one the authors can employ to continuously deepen connections with each other and Black Canadian participants in different regions, either in-person or online. The team started with a first wave selection by asking at least two nonwhite Deaf/non-Deaf people to help us to identify Black Deaf Canadians in Nova Scotia. In total, three Black Deaf/non-Deaf people responded and contacted us.

The team engaged in individual conversations with these three Black participants—two Deaf and one hearing. At the time of writing, one participant had volunteered to help the team to recruit participants. With support from the team and participants, this study may gain a few new participants to be involved in dialogues within the next year. The more data we have, the richer the information that the team can analyze and share with the public will be.

### 6.3. Conversations

We sought informed consent from our participants—two local Black Deaf participants at their respective private homes and one Black hearing participant at an archival site. Three members of the team each had four semistructured questions to engage the participants in dialogue, while a fourth member recorded the conversation with one camcorder and one digital camera. The questions served as prompts for informal dialogue in the Black Deaf (and Black hearing) participants' Sign Language.

During the conversations, participants brought materials to share with the team, such as photos of classmates and extracurricular activities at the schools they had attended. One participant used their electronic device to show their history and interests (e.g., stamp collection).

### 6.4. Transcripts (In Process)

As seen in Figure 2, a member of the team reviewed and transcribed the recorded videos of each dialogue. That is, the member translated between a signer, ASL Gloss (verbatim), and plain English. A duplicate second transcript of plain English, as translated from Sign Language, was developed into ASL Gloss to allow for clear pictures of the members of the research team and the participants' representative cultural behavior and language use. Auto voice-to-text was used for transcription in exceptional cases when interpreting services were not available due to prohibitive costs or a limited availability of culturally appropriate interpreters (see Figure 3).

In the coming months, team members will code for recurring themes or topics as discussed from the conversations (Cohen et al. 2011). In greater detail, the practice of axial coding (open coding) with the generation of core categories (theoretical coding) may enable the team to capture, identify, and explain overall impressions and different behaviors, appearances, and Sign Languages. The collection and analysis of the data will reveal informative narratives of historical and contemporary contexts from Black Deaf individuals' perspectives and experiences.

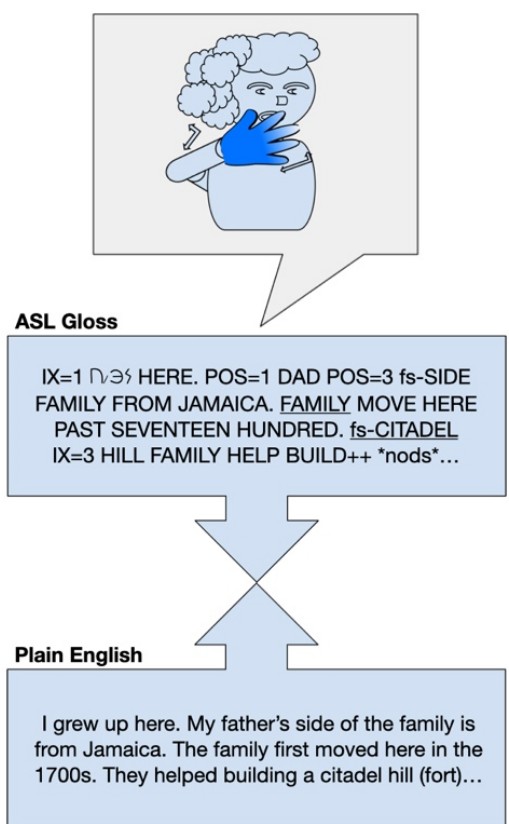

**ASL Gloss**

IX=1 ∩∋ʃ HERE. POS=1 DAD POS=3 fs-SIDE
FAMILY FROM JAMAICA. <u>FAMILY</u> MOVE HERE
PAST SEVENTEEN HUNDRED. <u>fs-CITADEL</u>
IX=3 HILL FAMILY HELP BUILD++ *nods*...

**Plain English**

I grew up here. My father's side of the family is
from Jamaica. The family first moved here in the
1700s. They helped building a citadel hill (fort)...

**Figure 2.** This is a sample of how Participant 1 was signing in a conversation, as captured from video recordings that were manually transcribed into American Sign Language (ASL) Gloss. Graphemes (i.e., symbols) are included to document any unfamiliar or unusual sign productions. For example, Participant 1 produced a word in a different palm orientation for "GROW > UP". The second box illustrates how we manually transcribed live conversations into plain English to the greatest extent possible.

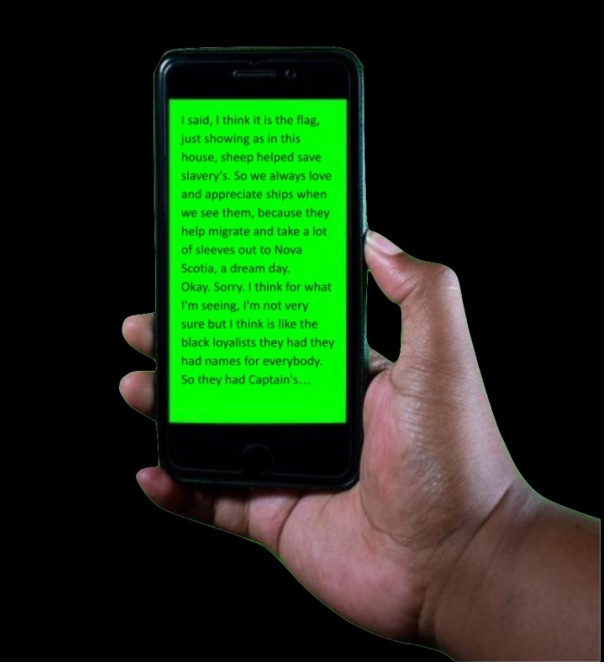

**Figure 3.** This is a separate sample showing spoken conversation from a tour at a museum, as transcribed by a phone's voice-to-text app. Word correction examples: sheep—ship; sleeves—slaves.

One of the many learnings from the research trip is that research requires time and patience; there is more for us to find, study, and analyze. We initially wanted to talk to at least five local Black Deaf and Black hearing participants at a social gathering (either private or public, such as a Black museum, community center, library, or park), but we ended up with only three participants. Another lesson came from the fact that during every hour-long dialogue, the battery of the camcorder ran out. This experience taught us that the research equipment needs to meet the length of our dialogues, and perhaps a professional videographer needs to be hired. A third lesson was about the relationships within the team. For some members, the research process was a new and challenging experience. The consultation with other team members throughout the process helped to deepen relationships that facilitated clearer communication. Relationship and trust building prior to the trip was helpful for inspiring that process.

## 7. Research Team: During the Research Trip

It is essential for the reader to know about how Canada was formed to contextualize the research trip and its findings. During the years between 1850 and 1861, Canada was not an official country as we know it today. For the decade between 1860 and 1870, Canada had two parts: West and East.

The narratives of Africville, Nova Scotia have attracted the attention of scholars and historians of all stripes. It is commonly known that, after the first Black settlement in 1794, the intergenerational community of Africville experienced racial oppression that continued for centuries. Black individuals were forced to live on uninhabitable land under difficult conditions (Clifford [1999] 2006; Nelson 2008; Rutland 2011). Despite the hardships, they managed to form a strong sense of community and thrive without governmental support (Hill 2007; Nelson 2008; Rutland 2011).

Based on the team's preliminary research findings from dialogues and archival sites in Nova Scotia, Prince Edward Island, and New Brunswick, there were several other active Black communities in East Coast cities, outside of Africville, Nova Scotia. For example, there were Black communities in Shelburne, East Preston, Annapolis Royal, Cherry Brook, Halifax, Sydney, Springhill, North Preston, and Beechville. Some Black Deaf Canadians from intergenerational Black communities within thirty kilometers of Halifax attended a school for the Deaf, which was built in 1856 on Gottigen Street (it was demolished and rebuilt in 1896 for a growing number of Deaf student enrolments from the Maritimes. The school of the Deaf in Halifax was in operation from 1896 to 1961 before it was relocated to Amherst, Nova Scotia; see (Carbin 1996)).

The terms Black Deaf, or even Deaf, rarely appeared in online archives. Instead, they are found in archival and museum sites' primary sources. The terms used by historians, surveyors, and other data collectors to identify or describe Black Deaf individuals include "Negro" and "colored". It was also a challenge to find specific and credible information on Black Deaf Canadian history, because information such as Black Deaf excellence in Canada in the distant past is ephemeral (short-lived). For example, a newspaper article in the early 20th century briefly shared a minister's recollection of a quiet Black Deaf child. The team unsuccessfully tried to find more information on the child, even with the archivists' help. A variety of documents (such as studies, reports, and censuses) typically use deficit terminology such as "deaf and dumb", "people who cannot hear", "hearing impaired", and "hearing loss". While the first term is now widely recognized to be offensive, the last three terms are still being used in audiocentric communities. The stigmatization connected to this deficit terminology further deters Black Deaf Canadians from knowing their history and from embracing their identities, languages, and other aspects of their lives.

## 8. Relat(ing) Identity Insights

Upon returning home from the three-week trip, we discussed and observed how we, as Black Deaf Canadians, support each other amidst community attitudes rooted in ableism. When we meticulously searched for a meaningful (re)construction of self through,

for example, statistics, censuses, materials (e.g., photos), and dialogues, we uncovered unexpected information. In the past, Black Deaf Canadians experienced limited (shortened) education. They were offered prolonged and repetitive vocational training that led to the early onset of additional disabilities, underemployment, and a lack of socialization with Black hearing communities. This corresponds with the documented history of education in the United States for Black Deaf children, including a lack of access to sign language, low expectations, poorly funded resources, and prioritization of education advancement (McCaskill et al. 2011, 2022; Simms and Thumann 2007; Simms et al. 2008), and reduced livelihoods for Black Deaf adults due to the low diversity of professions, insufficient training, and medical conditions (McCaskill et al. 2011, 2022; Perrodin-Njoku et al. 2022).

Despite all of this, Black Deaf Canadians maintained optimal attitudes and strong relationships with their children. For example, a Black Deaf research participant shared that they went to a predominantly white mainstream school as a child, and their dream of becoming a judge was denied outright because they were Black Deaf. The participant was instead redirected to take vocational training when their teachers would not provide resources to ensure that their academic learning and Sign Language needs were met. The participant was whitewashed to the point where they used to believe that no Black Deaf individuals were capable of advancing their education. Not only were their future social and economic capacities suppressed by vocational education, but so were their Black Deaf identities. This participant eventually acquired Sign Language, and it is through Sign Language that they communicated their dreams to their children. Sign language is valuable for communicating dreams and, in doing so, (re)constructs history and identity. All of this participant's children are now studying human rights law, especially concerning Black Deaf lives.

## 9. Concluding Considerations

Black Deaf Canada and the Sign Language Deaf community are grappling with a profound lack of resources, knowledge, and opportunities. We need resources to make Black Deaf Canadian's histories visible and to foster positive cultural identities. This is especially important in the wake of the latest iteration of the Black Lives Matter movement and in the context of overt government and system-sanctioned racism.

This article shows how the authors overcame ableism and audism integrated with racism by learning, supporting, and working together. Our relationships led to the enrichment of the research project. There has been steady growth in the documentation and scholarship of Black Deaf Canadians through the application of a combined framework–Moges' (2020) Black Deaf Gain, and Annamma et al.'s (2016) DisCrit. The more people we met during our research trip, the more enlightened we became regarding Black Deaf history and identity formation. Ongoing relationship-building and decision-making helped to (re)construct Black community history and to redefine Black Deaf Canadian relatable identity/ies, changing them/us from invisible to visible. The project moved us from being buried in audism and ableism to building the community and deepening the understanding of the diversity of Black Deaf experiences in Canada. Community-building occurred among the research team, with Black Deaf Canadians, and with the museum, library, and archive personnel who we met and educated in the process of our research.

Black Deaf Canadians will benefit from gaining knowledge and community-building through our growing historical narratives. Importantly, our action research enables us to transform our identities as individual Black Deaf Canadians prior to our organization to form a collective and shared identity as we continue our research and relationship-building. Building upon the hidden history and understanding of the Black Deaf community is essential, noteworthy, and invaluable, especially when carried out with/in Black, Black Deaf, and Black disability communities.

Academic study of Deaf experiences and language has primarily focused on the white Deaf experience. Meanwhile, the academic study of Black Canadian communities has solely focused on racial experiences and not broader disabled experiences. Narratives of Black

Deaf Canadian experiences are under-documented and under-represented. Our experiences often fade into the distant past or the invisible present. Despite the omission of Black Deaf Canadians from reports, statistics, and censuses, a few mentions are documented in books, as discussed. As such, through Black Deaf Canada, we are discovering and reconstructing our history/ies.

Currently, the experience of audism is closely interwoven with ableism, racism, and other experiences of disability in Black hearing, nonsigning communities. This is a reminder of the glaring gap in the histories of Black Deaf Canadians that the research team is addressing. Our ways of tackling this gap are deeply personal, rooted in community, and challenge the status quo.

The preliminary findings from this research project provide some answers to Black Deaf Canadians of all ages. We continue to collect data, plan future research trips, and seek Black Deaf Canadians to engage in dialogue. We look forward to the analysis of the coded data from our transcripts. Despite layered oppression, narratives of Black Deaf Canadians' lived experiences, stories, and connections do live on, and—as we are discovering—are identifiable, tangible, and relatable.

While visible and/or tangible representation matters to us, identity is a vital part of the Black Deaf community because of a shared struggle to create a safe space for ourselves. Narratives of past, present, and future generations of Black Deaf Canadians are what we are fighting for: more visibility; more representation; more documentation; no more language shaming. We are centering and elevating our Black Deaf Canadian experiences in the pursuit of better education, greater access, and comprehensive resources. We broadcast with pride: We see ourselves in historical and contemporary Black Deaf Canadian narratives. Now, we know where and how to find Black Deaf Canadians in communities, in documentation, and in records. *We can finally relate.*

**Author Contributions:** Conceptualization, J.R.; methodology, J.R. and A.P. (Amy Parsons); validation, J.R.; writing—original draft preparation, J.R., A.P. (Amelia Palmer) and A.P. (Amy Parsons); writing—review and editing, J.R., A.P. (Amelia Palmer) and A.P. (Amy Parsons); visualization, J.R.; supervision, J.R.; project administration, J.R., A.P. (Amelia Palmer) and A.P. (Amy Parsons). All authors have read and agreed to the published version of the manuscript.

**Funding:** This research received no external funding.

**Institutional Review Board Statement:** Not applicable. The nontraditional independent research project belongs to Black Deaf Canada, as discussed in this article.

**Informed Consent Statement:** Informed consent was obtained from all subjects involved in the study.

**Data Availability Statement:** Data is not available due to privacy.

**Conflicts of Interest:** The authors declare no conflict of interest.

## Notes

[1] Considering intersectionality theory explored from two standpoints of racialized people—one hearing (Hill Collins 2000) and another deaf (Eckert and Rowley 2013), our research team has discussed and intentionally chosen to use the term, "Black Deaf", as it reflects both our individual and collective lived experiences.

[2] According to McCaskill et al. (2011), racial segregation in its modern form began in the late 1800s in the United States where slavery had existed for more than two hundred years prior to the Civil War. Racial segregation resulted in public policy that banned Black people from a higher standard of education.

[3] Although Black Deaf Canadian experience and narrative is omitted from the reports, statistics and censuses, it is acknowledged in a small number of documents. Examples include: McAskill et al.'s (2022) theatre arts article, "Interview with Natasha Bacchus, a.k.a. Courage", Clifton F. Carbin's (1996) years-long documented history book, *Deaf Heritage in Canada: A Distinctive Diverse and Enduring Culture*, and Evelyne Gounetenzi's (1999) hard copy report, Multiculturalism and the Deaf".

[4] Eckert and Rowley (2013) further explain contextual audism in relation to discrimination of overt, covert, and aversive behaviors.

[5] The remainder of responses included 17 percent reporting their experience as somewhat negative; 21 percent as somewhat positive, and 21 percent as mostly positive (Parsons 2022b).

[6] This hearing activist person asked to remain anonymous.

7    We use the word "relat(ing)" instead of "related" to illustrate a continuous action of uncovering and relearning history that in turn evolves our identities as Black Deaf Canadians in a non-static manner—whether the history itself is distant, recent, or current.

8    Danielle Pethebridge (2016) discusses vulnerability in a critical framework to understand individuals' individuality-related roles in various contexts.

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
