# Peer review of "Reconstruct(ing) a Hidden History: Black Deaf Canadian Relat(ing) Identity"

_socsci, doi:10.3390/socsci12050305_

Round 1

Reviewer 1 Report

This paper addresses critical issues regarding Black Deaf Canadians who have been historically excluded from dominant deaf community narratives. It discusses important aspects of intersectionality (i.e., how Black Deaf and disabled people relate to Black communities). I really like the points at the top of p. 4 about Black disability research. I appreciate the description of the founding of a club and then organization for Black Deaf Canadians and the community-building this involved. I also like the focus on archival research. 

I am wondering about the authors’ non-use of Black ASL (the term used by McCaskill et al. and Hill who carried out sociolinguistic research in this area). I wonder if the authors can mention the history behind McCaskill and Hill’s research regarding the sign language varieties that emerged in Southern US segregated schools for Black Deaf children (after the Civil War and until the Brown v. Board of Education decision that ordered desegregation). 

What about using Black Canadian ASL/LSQ if you feel this fits?

-p. 2 near top: I feel the concept of Deaf Gain (Bauman & Murray) is not clearly described. Bauman & Murray write that they “coined the term of Deaf Gain to counter the frame of hearing loss as it refers to the unique cognitive, creative, and cultural gains manifested through deaf ways of being in the world” (p. xv).

I feel Moges does not focus on Black Deaf people as a “biologically diverse group” but on the social and historical experiences of Black Deaf people. I suggest not using the term “biologically diverse” because race is not a biological attribute (as W.E.B. Du Bois argued). 

-on p. 4 and elsewhere, referring to Richard Clark Eckert (2010): Note Dr. Eckert is American Indian (Ojibwe). 

I am wondering about using “woman” instead of “female” when referring to Chapple’s work? This is because “female” is a binary term.

-p. 5: I am wondering about the reference to “congenital disabilities” since not all deaf people are born deaf. Are you excluding deaf people who become deaf in childhood and learn sign language?

-p. 7: I really like the biographies of research team members. Maybe when the article is published, names of team members/authors could be shared? 

-p. 10: One thought I had is that sometimes, research participants like the people interviewed here want to be named in research. Maybe this is something to be considered in the future.

p. 11: I really like the way the authors’ different data is featured (ASL gloss, English translation, speech-to-text).

-p. 12: I am wondering about naming Nova Scotia instead of referring to “an eastern province.”

The information discovered about how Black Deaf Canadians were not encouraged to pursue education corresponds with the history of education for African-American Deaf children (lower funding and fewer resources were provided for Black Deaf schools in the USA). I think Carolyn McCaskill’s Ph.D. dissertation mentions this.

Author Response

Thank you for your time to review our manuscript - here are our responses in bold next to your comments below:

I am wondering about the authors’ non-use of Black ASL (the term used by McCaskill et al. and Hill who carried out sociolinguistic research in this area). I wonder if the authors can mention the history behind McCaskill and Hill’s research regarding the sign language varieties that emerged in Southern US segregated schools for Black Deaf children (after the Civil War and until the Brown v. Board of Education decision that ordered desegregation).  Done - it can be found in page 2, 3rd para. Thank you!

What about using Black Canadian ASL/LSQ if you feel this fits?  - we discussed, reviewed your suggested readings, and revised. 

-p. 2 near top: I feel the concept of Deaf Gain (Bauman & Murray) is not clearly described. Bauman & Murray write that they “coined the term of Deaf Gain to counter the frame of hearing loss as it refers to the unique cognitive, creative, and cultural gains manifested through deaf ways of being in the world” (p. xv). - we corrected it accordingly; thank you

I feel Moges does not focus on Black Deaf people as a “biologically diverse group” but on the social and historical experiences of Black Deaf people. I suggest not using the term “biologically diverse” because race is not a biological attribute (as W.E.B. Du Bois argued). - noted and removed

-on p. 4 and elsewhere, referring to Richard Clark Eckert (2010): Note Dr. Eckert is American Indian (Ojibwe). - revised

I am wondering about using “woman” instead of “female” when referring to Chapple’s work? This is because “female” is a binary term. - we corrected the term

-p. 5: I am wondering about the reference to “congenital disabilities” since not all deaf people are born deaf. Are you excluding deaf people who become deaf in childhood and learn sign language? - Updated by adding non-congenital; revised

-p. 7: I really like the biographies of research team members. Maybe when the article is published, names of team members/authors could be shared? -Excellent! Probably yes, if they allow us to do so!

-p. 10: One thought I had is that sometimes, research participants like the people interviewed here want to be named in research. Maybe this is something to be considered in the future. - we agree with you; we already did consider this for our continued data collection...

p. 11: I really like the way the authors’ different data is featured (ASL gloss, English translation, speech-to-text). - excellent! Thank you

-p. 12: I am wondering about naming Nova Scotia instead of referring to “an eastern province.”  - revised

The information discovered about how Black Deaf Canadians were not encouraged to pursue education corresponds with the history of education for African-American Deaf children (lower funding and fewer resources were provided for Black Deaf schools in the USA). I think Carolyn McCaskill’s Ph.D. dissertation mentions this. - we agree with you. We also included Dr. Laurene Simms and her team's observation in the paper.

Reviewer 2 Report

This article is a rich articulation of Black Deaf creation and collaboration that points to the difficulties of surfacing Black Deaf Canadian histories, and to the creative, archival, and community-based methodologies that this group of authors is exploring as they grow Black Deaf narratives. A particularly important contribution of this work is the author’s critical points about the whiteness of Deaf scholarship and the binary this creates (Black or Deaf), which has a lasting impact on the ways in which histories are determined, sought out, and uncovered.

Going section by section, there are many part of this article I appreciated, and in some moments I wondered if the authors might expand further. For example, I appreciated Section 4, “Research Team: Relationship-Building and Decision-Making Process” because it offered insight into the narrators and their perspectives—this is a nice anchoring of subjectivity in the overall article. However, Section 5, “Emergence of the Research Report” introduces the topic of vulnerability. Given that vulnerability is a richly theorized concept (Judith Butler’s work, for example), there is more room to deepen the analysis of vulnerability here. I am not aware of intersections in the literature between Deaf perspectives and vulnerability, but if the authors can identify or sculpt this intersection it would strengthen the paper and give more weight to the Black Deaf perspectives they assert (Sami Shalk's latest work might also be valuable here).

Additionally, in Section 6 “Method,” I wonder if the authors can further develop their analysis of archival methods in relation to the special issue’s call toward reworlding and relationship/community building (which are themes that emerge later). Although the authors fairly tell readers that they are new to archival methodologies, such methodologies are political and deeply invested in the preservation of existent worlds and the production of new ones—albeit in ways rife with incompleteness and, at times, patchwork data. What’s more, participating in archival methods, such as self-toured visits on a street to find where a school for the deaf used to be, seem to be significant acts of reworlding. I’m wondering (hoping) that the authors are constructing (or contributing to) a praxis of Deaf Black reworlding through archival methods, and that they can tell readers about this work. (Given that the terms “Black Deaf” and “Deaf” were not easy to find in some archival spaces, is it possible that the authors are creating research experiences that will search-able in this way in the future? An exciting prospect!)

Finally: I understand that there is a grammatical difference between the languages of English and ASL. As an English speaker, I am consistently noticing grammatical errors in the piece. I wonder, though, if I am reading incorrectly. Perhaps I should be reading with consideration for ASL’s grammar and structure—indeed if we imagine this article signed, there may be no errors to resolve. I suggest that one of two things happen: if the authors wish to present an English article, the text should be copy-edited for grammar; or, if the authors wish to present the written translation of ASL, this should be noted somewhere so that readers don’t interpret unfamiliar grammar as error. While I realize the latter is a very English-centric suggestion, I put it forward with the hope of guarding against audiocentric academic language shaming. (Notably, Figures 2.1 and 2.2. are significant to showing readers how participants were signing, and reminding readers that signing is embedded in this work).

Here are some suggestions for clarity for the English reader:

1.       Suggestion to change phrasing from “Senseless lives lost” to “lives senselessly lost.”

2.       While I understand that this acronym is well known, please consider spelling out “BLM” on first mention for clarity.

3.       If possible, please name the Black civil rights activist (who never sat upstairs in a theatre) to further enhance the significant of this story.

4.       Consider rephrasing the first line of Section 9 “Concluding Considerations.” The phrase “Although living in the country” implies that the researcher are living on the countryside or in a rural area, which I don’t think is the intended meaning. A suggestion might be, “Although based in a country with Black Deaf Canadian histories, our organization is a Sign Language Deaf community that is grappling with a deep lack of resources, knowledge, and opportunities to make these histories visible.” (I’m not sure if this suggestion captures your meaning correctly, please disregard if not).

Author Response

Thank you for your time to review our manuscript - here are our responses in bold next to your comments below:

Going section by section, there are many part of this article I appreciated, and in some moments I wondered if the authors might expand further. For example, I appreciated Section 4, “Research Team: Relationship-Building and Decision-Making Process” because it offered insight into the narrators and their perspectives—this is a nice anchoring of subjectivity in the overall article. However, Section 5, “Emergence of the Research Report” introduces the topic of vulnerability. Given that vulnerability is a richly theorized concept (Judith Butler’s work, for example), there is more room to deepen the analysis of vulnerability here. I am not aware of intersections in the literature between Deaf perspectives and vulnerability, but if the authors can identify or sculpt this intersection it would strengthen the paper and give more weight to the Black Deaf perspectives they assert (Sami Shalk's latest work might also be valuable here). -Noted, reviewed, and revised.

Additionally, in Section 6 “Method,” I wonder if the authors can further develop their analysis of archival methods in relation to the special issue’s call toward reworlding and relationship/community building (which are themes that emerge later). Although the authors fairly tell readers that they are new to archival methodologies, such methodologies are political and deeply invested in the preservation of existent worlds and the production of new ones—albeit in ways rife with incompleteness and, at times, patchwork data. What’s more, participating in archival methods, such as self-toured visits on a street to find where a school for the deaf used to be, seem to be significant acts of reworlding. I’m wondering (hoping) that the authors are constructing (or contributing to) a praxis of Deaf Black reworlding through archival methods, and that they can tell readers about this work. (Given that the terms “Black Deaf” and “Deaf” were not easy to find in some archival spaces, is it possible that the authors are creating research experiences that will search-able in this way in the future? An exciting prospect!) -Noted, reviewed, and revised.

Finally: I understand that there is a grammatical difference between the languages of English and ASL. - Yes, that is correct. A slightly difference between these languages influence a significant meaning, indeed.

As an English speaker, I am consistently noticing grammatical errors in the piece. I wonder, though, if I am reading incorrectly. Perhaps I should be reading with consideration for ASL’s grammar and structure—indeed if we imagine this article signed, there may be no errors to resolve. I suggest that one of two things happen: if the authors wish to present an English article, the text should be copy-edited for grammar; or, if the authors wish to present the written translation of ASL, this should be noted somewhere so that readers don’t interpret unfamiliar grammar as error. While I realize the latter is a very English-centric suggestion, I put it forward with the hope of guarding against audiocentric academic language shaming. (Notably, Figures 2.1 and 2.2. are significant to showing readers how participants were signing, and reminding readers that signing is embedded in this work). - we like your second suggestion to strongly encourage the English reader to accept the changes and be aware of the presence of Sign Language on a daily basis if ever. This approach can be used for future articles. For now, a native English-speaking colleague reviewed it for grammars - thank you!

Here are some suggestions for clarity for the English reader:

1.       Suggestion to change phrasing from “Senseless lives lost” to “lives senselessly lost.” - revised

2.       While I understand that this acronym is well known, please consider spelling out “BLM” on first mention for clarity. - revised

3.       If possible, please name the Black civil rights activist (who never sat upstairs in a theatre) to further enhance the significant of this story. -accepted—the name was mentioned in various published newspaper and books. Revised

4.       Consider rephrasing the first line of Section 9 “Concluding Considerations.” The phrase “Although living in the country” implies that the researcher are living on the countryside or in a rural area, which I don’t think is the intended meaning. A suggestion might be, “Although based in a country with Black Deaf Canadian histories, our organization is a Sign Language Deaf community that is grappling with a deep lack of resources, knowledge, and opportunities to make these histories visible.” (I’m not sure if this suggestion captures your meaning correctly, please disregard if not).  - revised, thank you!